# Conscious Sedation for Dental Treatments in Subjects with Intellectual Disability: A Systematic Review and Meta-Analysis

**DOI:** 10.3390/ijerph20031779

**Published:** 2023-01-18

**Authors:** Claudia Salerno, Silvia Cirio, Giulia Zambon, Valeria D’Avola, Roberta Gaia Parcianello, Cinzia Maspero, Guglielmo Campus, Maria Grazia Cagetti

**Affiliations:** 1Department of Biomedical, Surgical and Dental Sciences, University of Milan, Via Beldiletto 1, 20142 Milan, Italy; 2Fondazione IRCCS Cà Granda, Ospedale Maggiore Policlinico, University of Milan, 20100 Milan, Italy; 3Department of Restorative, Preventive and Pediatric Dentistry, University of Bern, Freiburgstrasse 7, 3012 Bern, Switzerland

**Keywords:** conscious sedation, intellectual disability, special need patients, dental treatment

## Abstract

This systematic review and meta-analysis was aimed to investigate the conscious sedation efficiency in patients with intellectual disability undergoing dental treatment (PROSPERO CRD42022344292). Four scientific databases were searched by ad-hoc prepared strings. The literature search yielded 731 papers: 426 were selected, 42 were obtained in full-text format, and 4 more were added after hand searching. Fourteen studies were finally included, 11 of which were included in the meta-analysis (random effect model). A high heterogeneity in the drugs used and route of administration was retrieved. Success rate, occurrence of side effects, and deep sedation occurrence were combined to give an overall efficiency of each drug. N_2_O/O_2_ reported the highest efficiency (effect size = 0.90; *p* < 0.01) and proved to be more efficient when used alone. Nine papers reported a success rate of sedation of 80% or more. The prevalence of side effects (6 studies) ranged from 3% to 40%. Enteral and parenteral benzodiazepines showed the same overall efficiency (effect size = 0.86). No meta-analysis has yet been conducted to define the most effective and safest way to achieve conscious sedation in patients with intellectual disability; nitrous oxide appears to be the best choice to perform conscious sedation in patients with intellectual disability undergoing dental treatment.

## 1. Introduction

The dental management of patients with intellectual disability (ID) presents a great challenge for the clinician. Individuals with intellectual disability often encounter difficulties in interacting with the physician and tend to refuse treatment because of cognitive impairment; additionally, they may exhibit high levels of anxiety related to medical examinations and procedures [1]. Children with ID have usually higher caries prevalence with respect to the general population [2,3]. Adults with ID have poorer oral hygiene and higher prevalence and severity of periodontal disease; conversely, their caries rates do not seem to differ from those of the general population [4,5]. However, the overall rates of untreated caries are consistently high in people with ID [6]. An urgent call for the development of strategies to increase the acceptance of both preventive and therapeutic dental procedures and to facilitate clinicians in providing them is needed. Increasing dental procedure acceptance requires highly personalized approaches that include behavioral and pharmacological strategies [6].

Still, the most common approach for providing dental care to patients with intellectual impairment is general anesthesia (GA). Indeed, an overuse of deep sedation techniques and general anesthesia has been reported for individuals with moderate to severe intellectual disability, for whom GA is used in up to 60% of cases [7]. However, GA presents significant risks in terms of morbidity and mortality, as well as discomfort for patients and their families, and last but not least, substantial costs for public healthcare or families. General anesthesia should be limited to situations in which alternative approaches are not feasible [8,9]. According to the American Academy of Pediatric Dentistry, effective alternatives to GA in uncooperative patients and those with disability include behavioral management techniques, protective stabilization, and conscious sedation [10]. Conscious sedation is a technique in which the use of gases or/and drugs, not requiring the professional presence of an anesthetist, produces a state of depression of the central nervous system that allows the patient to remain conscious, to maintains protective reflexes independently and continuously, with the ability to respond appropriately to physical stimulation and/or verbal command [11,12]. This procedure is safe and suitable for outpatient settings, and it can be performed with the administration of different drugs and routes [13]. Sedation is mainly performed with benzodiazepines, administered orally or intravenously, or with nitrous oxide (N_2_O) by inhalation. Benzodiazepines reduce anxiety and aggressive behavior and induce muscle relaxation; an anticonvulsant effect is also produced [14]. Nitrous oxide is an anesthetic gas that causes central nervous system depression with an effective anxiolytic effect and a mild analgesic effect at sub-anesthetic concentrations [15].

In order to perform dental treatments in uncooperative patients, N_2_O and benzodiazepines can be used separately or in combination, depending on the degree of cooperation offered, the severity of the intellectual disability, and the complexity of the dental treatment to be performed [16]. Other drugs described in the literature are meperidine and promethazine, administered orally, or low-dose intravenous propofol [17,18,19].

No scientific evidence or clinical recommendation on the dosage and type of drug to be used in the conscious sedation of individuals with intellectual disability undergoing invasive dental treatment is available. Based on this premise, the aim of the present systematic review and meta-analysis is to fill this gap by analyzing and comparing clinical studies that have investigated the effectiveness of drug- or gas-mediated conscious sedation in patients with intellectual disabilities undergoing dental treatment.

## 2. Materials and Methods

The present systematic review was registered a priori in the International Prospective Register of Systematic Reviews (PROSPERO) under protocol number CRD42022344292. This review has been conducted and reported according to the Cochrane Handbook of Systematic Reviews of Interventions [20] and the guidelines of Preferred Reporting Items for Systematic Review and Meta-Analysis (PRISMA) [21]. The PRISMA checklist is displayed in Appendix A.

The PICO model was used to structure the clinical research question by defining the inclusion criteria [22]. Thus, the present review aimed to systematically search and analyze clinical studies investigating which drug/agent, in terms of sedative efficacy, behavioral control, and safety should be used in subjects with intellectual disability undergoing dental treatment and which route of administration is the most effective and preferred.

Population: subjects with intellectual disability in need of dental treatment.Intervention: drug- or gas-mediated conscious sedation performed for dental treatment.Comparison: no comparison, no drug intervention, different drugs comparison or different dosages.Outcome: sedative, behavioral, anxiolytic efficacy, and safety of the sedative interventions.-Sedative efficacy. Measurement scales include the Ramsay Sedation Scale (scored 1–6 based on the response of the patient), the Richmond Agitation Sedation Scale (scores from +4 to −5), the State Behavioral Scale (scored from −3 to +2), the Bispectral Index Monitoring (range of scores from 0 to 100) and the Classification of Emotional Status designed by Breitkopf and Buttner (scored from 1–4).-Behavioral efficacy. Measurement scales include the Houpt Behavior Rating Scale (subdivided into 4 scales, each sub-scale then defined separately), the Frankl Behavior Rating Scale, the FLACC, the Venham Scale (scores from 0–3), the Visual Analogue Scale (scale from 0–10).-Anxiolytic efficacy. Measurement scales include pulse rate, the Children’s Fear Survey Schedule Dental Subscale (defined by 15 scores based on the item that the child is fearful of), and the Spielberger State Anxiety Inventory (psychological inventory based on a 4-point Likert scale consisting of 40 questions).-Safety. Evaluation includes side effects defined as any undesired harmful effects or reactions to the sedative agents during or after administration.

### 2.1. Eligibility Criteria

The inclusion criteria were:Type of study: all types of clinical studies except for case series or case studies;Publication languages: papers published in English, Italian, and French;Time of publication: no time restrictions were applied;Type of intervention: conscious sedation with N_2_O or sedative drugs in subjects with intellectual disabilities undergoing dental treatments;Outcomes: sedative, behavioral, anxiolytic effectiveness, and safety of the intervention used to improve collaboration during dental treatments.The exclusion criteria were:Studies for which the full text is not available.

### 2.2. Information Sources and Search Strategy

Four electronic databases, PubMed, Embase, Scopus, and Cochrane Central Register of Controlled Trials, were searched from inception until 1 September 2022 by two authors (C.S. and S.C.). The search strategy included a search string for each database:For PubMed, the string used was: (“Neurodevelopmental Disorders” [Mesh] OR “Disabled Persons” [Mesh] OR special needs) AND (“Dentistry” [Mesh] OR “Oral Health” [Mesh] OR “Mouth” [Mesh] OR “Dental Health Services” [Mesh] OR “Dent*” [Title/Abstract]) AND (“Dental Anxiety” [MeSH Terms] OR Behavior*[Title/Abstract] OR Collaboration[Title/Abstract] OR Succes*[Title/Abstract]) AND (“Benzodiazepines” [Mesh] OR “Tranquilizing Agents” [Pharmacological Action] OR “Tranquilizing Agents” [Mesh] OR “Imidazoles” [Mesh] OR “nitrous oxide” OR ketamine OR “Psychotropic Drugs” [Mesh] OR “conscious sedation” OR n2o OR “moderate sedation” OR “mild sedation”).For Embase: (‘mental disease’/exp OR ‘mental disease’ OR ‘disabled person’/exp OR ‘disabled person’) AND (‘dentistry’/exp OR ‘dentistry’ OR ‘mouth’/exp OR ‘mouth’ OR ‘dental health’/exp OR ‘dental health’ OR ‘stomatognathic system’/exp OR ‘stomatognathic system’) AND (‘dental anxiety’/exp OR ‘dental anxiety’ OR ‘collaboration’/exp OR ‘collaboration’ OR ‘treatment success’/exp OR ‘treatment success’) AND (‘benzodiazepine’/exp OR ‘benzodiazepine’ OR ‘tranquilizer’/exp OR ‘tranquilizer’ OR ‘imidazole derivative’/exp OR ‘imidazole derivative’ OR ‘imidazole’/exp OR ‘imidazole’ OR ‘nitrous oxide’/exp OR ‘nitrous oxide’ OR ‘ketamine’/exp OR ‘ketamine’ OR ‘psychotropic agent’/exp OR ‘psychotropic agent’ OR ‘conscious sedation’/exp OR ‘conscious sedation’ OR ‘benzodiazepine derivative’/exp/mj OR ‘anxiolytic agent’/exp/mj OR ketamine OR ‘nitrous oxide’ OR ‘psychotropic agent’/exp/mj OR n2o OR ‘conscious sedation’).For Scopus: (TITLE-ABS-KEY(mental disease) OR TITLE-ABS-KEY(disabled person) OR TITLE-ABS-KEY(neurodevelopmental disorders) OR TITLE-ABS-KEY(Disabled) OR TITLE-ABS-KEY(Special needs)) AND (TITLE-ABS-KEY(dentistry) OR TITLE-ABS-KEY(oral health) OR TITLE-ABS-KEY(mouth) OR TITLE-ABS-KEY(dental health) OR TITLE-ABS-KEY(dent*) OR TITLE-ABS-KEY(stomatognathic system)) AND (TITLE-ABS-KEY(dental anxiety) OR TITLE-ABS-KEY(behav*) OR TITLE-ABS-KEY(collaboration) OR TITLE-ABS-KEY(success*) OR TITLE-ABS-KEY(treatment success) OR TITLE-ABS-KEY(dental fear)) AND (TITLE-ABS-KEY(benzodiazepine) OR TITLE-ABS-KEY(tranquilizing agents) OR TITLE-ABS-KEY(tranquilizing drug) OR TITLE-ABS-KEY(imidazole) OR TITLE-ABS-KEY(nitrous oxide) OR TITLE-ABS-KEY(n2o) OR TITLE-ABS-KEY(ketamine) OR TITLE-ABS-KEY(psychotropic drugs) OR TITLE-ABS-KEY(conscious sedation) OR TITLE-ABS-KEY(moderate sedation) OR TITLE-ABS-KEY(mild sedation) OR TITLE-ABS-KEY(psychotropic agent)).For Cochrane: (neurodevelopmental disorders OR disabled person OR special needs OR mental disease OR retarded person OR handicap OR impaired person) AND (dentistry OR oral health OR mouth OR dental health OR stomatognathic system OR dent*) AND (dental anxiety OR behavior OR collaboration OR cooperation OR compliance OR treatment success OR success) AND (benzodiazepines OR tranquilizing agents OR imidazoles OR nitrous oxide OR ketamine OR psychotropic drugs OR psychotropic medications OR sedatives OR conscious sedation OR n2o OR moderate sedation OR mild sedation OR anxiolytic agents OR narcotics).

Cross-referencing was also performed using the reference lists of full-text articles and grey literature retrieved via opengrey.eu (http://www.opengrey.eu, accessed on 10 December 2022).

### 2.3. Study Selection

The outputs of the literature search were uploaded into a spreadsheet (Microsoft Excel^®^), and duplicates were removed. Two authors (S.C. and C.S.) independently examined all papers by title and abstract, and papers meeting the inclusion criteria were obtained in full-text format. The same authors assessed the selected papers to establish whether each paper should or should not be included in the systematic review. Disagreements were resolved through discussion and/or full-text analysis in doubtful cases. Where resolution was not possible, another author was consulted (M.G.C.).

### 2.4. Data Collection and Synthesis

Data collection and synthesis were independently performed by three authors (G.Z., R.P. and V.D.) using an ad hoc designed data extraction form (Appendix A), without masking the name of the journal, title, or authors. Numerical data were extracted and rounded up to two decimals; if this was not possible, data were extracted as they were reported in the papers. Data were reported only if they were related to subjects with intellectual disability. In studies whose sample did not include only patients with intellectual disability, and if it was not possible to extrapolate data related to the subgroup under study, data were not reported.

### 2.5. Outcome Variables

Primary outcomes for this review were: effectiveness of sedation assessed through the completion of dental treatment, number of adverse events, and maintaining the adequate level of sedation. Secondary outcomes were behavioral and anxiolytic efficacy and any other variable considered in the included studies. Studies whose sample did not include only patients with intellectual disabilities were included in the study only if it was possible to extrapolate data for subjects with intellectual disabilities for at least one primary outcome.

### 2.6. Risk of Bias Assessment and Quality Assessment

The risk of bias assessment was carried out independently by three reviewers (C.S., S.C. and M.G.C.) using the Cochrane Collaboration’s RoB 2 and RoB 1 tools for RCT and NRSI studies, respectively. The Excel tool for RoB 2 was used to input answers given to signaling questions and then an algorithm estimated the overall risk of the bias according to the results for each domain as: “low risk”, “some concerns” or “high risk”. Risk of bias plots were drawn using the Cochrane robvis web app [23]. The ROBINS-I tool was used to assess the risk of bias in non-randomized studies of intervention (NRSI) [24]. Authors answered signaling questions in each domain and then estimated the overall risk of bias as: “low”, “moderate”, “serious”, or “critical”.

For the quality assessment, a list of confounding domains and co-interventions was agreed upon, and they were identified as: severity of disability, age range, type of dental treatment performed, and number of drugs administered. The NIH Study Quality Assessment Tools were used according to each type of study [25]. Blinding is often difficult in such studies and was considered “not applicable” as well as the drop-out since none of the studies foresaw follow-up. The quality was considered: high, when all criteria were met or no more than 1 criterion was judged unclear; medium, if 2 criteria were judged unclear and the others were met, or if 1 criterion was not met and the others were met; low, if 3 or more criteria were judged unclear and the others were met, or if 2 criteria were not met and the others were met.

### 2.7. Data Analysis

Prometa3 Software (IDoStatistics, Cesena (FC), Italy) was used for the meta-analysis. Meta-analysis was performed if two or more studies included comparable subjects, interventions, and outcomes. The sample size together with the number of episodes of successful sedation, deep sedation, and side effects were extracted or calculated for each study, and for each outcome variable to be meta-analyzed, using the sedation episode as the analysis unit. Heterogeneity of effects among studies was assessed by means of Cochran’s test for heterogeneity, with a significance threshold of *p* < 0.1. The percentage of variability in the effect estimates due to heterogeneity rather than chance was calculated with the I2 statistic. Due to high heterogeneity, meta-analysis was undertaken using a random effects model. The results of each meta-analysis were graphically presented by the effect size of forest plots. Sub-groups analyses were performed, when feasible, comparing in the same plot the data of different drugs. Where the meta-analysis appeared inappropriate, the results of the included studies were not pooled, and a qualitative description of the included studies with supporting data was presented.

## 3. Results

The literature search yielded 731 papers: 426 were selected after removing duplicates, then 384 were excluded after evaluation of the title and abstract with a proportional agreement between reviewers of 0.89 with a Cohen’s k of 0.48 (Appendix A: Excluded articles and reason of exclusion after title and abstract evaluation); 42 articles were obtained in their full-text format, and 4 more were added after consulting reference lists (Appendix A: List of articles added after consulting reference lists). After full-text analysis, 32 articles were discarded (Appendix A: Excluded articles and reason for exclusion after full-text analysis). Therefore, a total of 14 studies (3.29% of the initial pool) were included in the systematic review, 11 of which were included in the meta-analysis (Figure 1). The proportionate agreement at this stage between reviewers was 0.82, with a Cohen’s k of 0.64.

Corresponding authors’ countries included France [26,27,28,29], the UK [30,31,32], Brazil [33,34], the United States [17], Canada [35], Italy [36], the Netherlands [18], and Thailand [37].

### 3.1. Studies Characteristics

The included 14 papers were published between 1980 and 2019: 10 articles [18,26,27,28,29,31,33,34,36,37] were published between 2000 and 2020 and 4 [17,30,32,35] were published before the 2000s. In terms of the types of studies, 7 were prospective studies [27,28,29,31,33,34,35], 2 were retrospective studies [17,18], 3 were randomized control studies [30,32,37], 1 was a non-randomized control study [26] and 1 was an observational study [36] (Table 1).

In nearly all of the included studies [17,18,26,27,28,29,30,31,32,33,34,36,37], dentist was the first operator and in eight with the support of one anesthetist [17,18,30,31,32,33,34,37]. In only one study, the first operators were dental hygienists supported by anesthetists [35].

All the included papers aimed to investigate the effectiveness of sedation, which was assessed as successful in completing dental treatments. Five studies [26,30,32,35,37] included a comparison group.

### 3.2. Samples

Sample sizes in the selected studies ranged from 13 [37] to 349 [29] participants, with 4 studies having a sample size greater than 100 ([18] n = 124, [31] n = 289, [28] n = 325, [29] n = 349). Eight papers included only patients with intellectual disabilities [18,29,30,31,32,34,35,37], and the other six papers included non-selective samples from which data on subjects with intellectual disabilities were extrapolated [17,26,27,28,33,36].

The target population of 4 studies was children or adolescent under the age of 18 years [17,32,36,37], of 2 studies was adults [18,31] and of the remaining 8 studies was both children and adults [26,27,28,29,30,33,34,35].

Six studies [26,27,29,32,34,37] reported the sex of participants, showing a higher prevalence of males (57.06%) in the total sample.

### 3.3. Sedative Interventions

A great heterogeneity in the drugs used and route of administration was found in the included studies. N_2_O was administered in 8 studies [17,26,27,28,29,30,36,37], midazolam was the main drug or premedication in 6 studies [26,31,32,33,34,37], diazepam in 3 studies [30,35,37], propofol in 1 study [18], and meperidine and promethazine in 1 study [17]. The studies that used nitrous oxide alone or in combination with other drugs are presented in Table 2, while those that did not use N_2_O are shown in Table 3.

Only 9 studies reported the type of dental procedures carried out during sedation [18,26,27,28,33,34,35,36,37]. The procedures on both primary and permanent teeth ranged from a simple oral examination to complex treatments such as oral surgery or prosthetic treatments. Only two studies, however, associated the success rate of sedation with the type of intervention in subjects with intellectual disability [26,37].

### 3.4. Primary Outcomes

Three primary outcomes were considered: effectiveness of sedation assessed through completion of dental treatment, occurrence of side effects, and deep sedation (Table 2 and Table 3).

All included studies reported the success rate of conscious sedation, with 9 papers [18,26,27,28,29,33,34,35,37] reporting a success rate of 80% or more.

In seven studies [17,18,26,28,29,31,32], the prevalence and type of side effects in patients with ID were reported. Prevalence ranged from 3% [17] to 37% [18]; the most frequent effects were: respiratory alterations [18,26,28,29,31], nausea and vomiting [26,28,29,31], neurological effects [26,28,29], vaso-vagal symptoms [18,28,29], behavioral alterations [26,28,29], paradoxical effect, prolonged recovery, urinary incontinence [31], bradycardia and tachycardia [18], prolonged sedation [32], and other not otherwise specified effects [28,31].

Deep sedation was reported for patients with ID in 11 studies [18,26,27,28,29,30,31,32,33,35,37], with episodes occurring in 6 studies [18,26,30,31,33,37].

### 3.5. Secondary Outcomes

Secondary outcomes include patient behavior during treatment, level of sedation, acceptance of treatment, behavior at recall visits, patient satisfaction, dentist satisfaction, vital signs, influence of repetition of conscious sedation, and role of the operator on the success of sedation, comparison between experts in sedation and non-expert dentists.

Behavior was measured according to the Venham Behavior Scale in 5 studies [26,27,28,29,36], according to the Visual Analogue Scale in 1 study [27], according to a modified version of the Houpt Scale in 1 study [37], according to the Frankl’s modified scale in 1 study [32], and other customized behavioral scales in 3 studies [30,33,35].

Level of sedation was assessed according to the Ramsay score in 1 study [26], according to the five-point Observer’s Assessment of Alertness/Sedation scale in another study [18] and according to the customized sedation scale in one study [17]. One study assessed both behavior and level of sedation according to the Dental Sedation Teachers Group (DSTG) scales [31]. Additional measures for vital signs were described in 7 studies [18,26,31,32,34,35,36], but not all reported data (oxygen saturation, heart rate, and systolic/diastolic blood pressure).

Treatment acceptance was reported in 2 studies [30,31], but one of them did not report data separately for patients with intellectual impairment [30]. The other [31] reported an acceptance rate for dental treatment of 90.5%. Patient and dentist satisfaction was analyzed in one study [27], but data were not separately reported for patients with intellectual disabilities.

The influence of repeated conscious sedation was evaluated in a study using the Venham scale [26], which showed a significant improvement in cooperation at different stages of the procedure (*p* < 0.01). Two studies [17,28] reported on the role of the operator on success. One [28] compared trainee and experienced dentists: trainees had statistically more failures in treating patients with intellectual disabilities (*p* < 0.01). The other [17] compared the gender of the practitioner, but the data were not reported separately for patients with intellectual impairment.

### 3.6. Risk of Bias and Quality Assessment

The two RCTs were both judged to have a moderate risk of bias [30,32]. The only non-randomized study [26] was judged to have a serious risk of bias, confounding domains and deviations from interventions that significantly affected the quality of rating (Figure 2).

The quality assessment of before-after studies with no control group, observational cohort studies and cross-sectional studies is shown in Figure 3. Four studies were judged to be of high quality [17,18,29,31], 5 were judged to be of medium quality [27,28,34,35,36] and one was of low quality [33]. Sample size and representativeness of sample were the two domains that most affected the quality of the studies (Figure 3).

### 3.7. Meta-Analysis

A random effects model was used to evaluate the success rate due to the high heterogeneity found (*p* < 0.01; I^2^ = 82.82%). Three studies [17,18,26] were excluded from the meta-analysis due to a lack of comparison studies. However, data on the success rate were the following: using N_2_O in addition to intravenous midazolam ES = 0.89, _95_CI (0.84/0.93) [26]; using N_2_O in addition to meperidine and promethazine ES = 0.68, _95_CI (0.57/0.74) [17], and finally, using intravenous propofol ES = 0.99, _95_CI (0.87/0.93) [18].

The studies included in the meta-analysis were grouped according to the drug used and the modality of administration into: enteral benzodiazepine (BDZ) (oral and rectal administration), N_2_O, N_2_O in addition to oral BDZ, and parenteral BDZ (intravenous and intranasal administration) (Figure 4). N_2_O/O_2_ reported the highest efficacy (ES = 0.84; *p* < 0.01) and proved to be more effective when used alone or together with midazolam than in combination with diazepam (Figure 4). Enteral BDZ and parenteral BDZ showed similar efficacy (ES = 0.76; ES = 0.77).

A random effects model was used to evaluate the absence of side effects due to the high heterogeneity found (*p* < 0.01; I^2^ = 95.07%). Studies excluded from the meta-analysis [17,18,26] reported: using N_2_O in addition to intravenous midazolam ES = 0.84, _95_CI (0.79/0.89) [26], using N_2_O in addition to meperidine and promethazine ES = 0.97, _95_CI (0.93/0.99) [17], and using intravenous propofol ES = 0.63, _95_CI (0.54/0.71) [18].

The studies included in the meta-analysis were grouped according to the drug used and the modality of administration into: enteral BDZ (oral administration), N_2_O, parenteral BDZ (intravenous and intranasal administration) (Figure 5). Parenteral BDZ reported the lowest number of adverse events (ES = 0.94; *p* < 0.01), while N_2_O and enteral BDZ showed similar results (ES = 0.83; ES = 0.84).

A random effects model was used to evaluate the non-occurrence of deep sedation due to the high heterogeneity found (*p* < 0.01; I^2^ = 88.83%). Studies excluded from the meta-analysis [18,26] showed the following results: using N_2_O in addition to intravenous midazolam, ES = 0.97, _95_CI (0.92/0.98) [26]; using intravenous propofol, ES = 0.73, _95_CI (0.64/0.80) [18].

The studies included in the meta-analysis were grouped according to the drug used and the modality of administration into: enteral BDZ (oral and rectal administration), N_2_O, N_2_O in addition to oral BDZ, and parenteral BDZ (intravenous and intranasal administration) (Figure 4). Enteral BDZ and N_2_O reported the highest safety (ES = 0.99; *p* < 0.01; ES = 0.98; *p* < 0.01); even N_2_O proved to be safe when used alone (Figure 6).

The three main outcomes (success rate, non-occurrence of side effects, and absence of deep sedation) were then combined (coefficient = 0.5) to give an overall efficiency of each drug. A random effects model was used to evaluate the success rate due to the high heterogeneity found (*p* < 0.01; I^2^ = 77.44%). Studies excluded from the meta-analysis [17,18,26] showed: using N_2_O in addition to intravenous midazolam, an overall success of ES = 0.91, _95_CI (0.87/0.94) [26], using N_2_O in addition to meperidine and promethazine, an overall success of ES = 0.90, 95CI (0.81/0.95) [17], using intravenous propofol, an overall success of ES = 0.88, _95_CI (0.78/0.94) [18].

The studies included in the meta-analysis were grouped according to the drug used and the modality of administration as follows: enteral BDZ (oral and rectal administration), N_2_O, N_2_O in addition to oral BDZ, and parenteral BDZ (intravenous and intranasal administration) (Figure 7). N_2_O reported the highest efficiency (ES = 0.90; *p* < 0.01) and proved to be more efficient when used alone (Figure 7). Enteral BDZ and parenteral BDZ showed the same overall efficiency (ES = 0.86).

## 4. Discussion

Anxiety control during dental procedures is essential to ensure the safety of the procedure, and to promote patient cooperation and patient and dentist satisfaction [38]. Psychological and pharmacological techniques are frequently used in dentistry, especially in patients with intellectual disability. Despite this premise, to the best of the authors’ knowledge, no meta-analysis has yet been conducted to define the most effective and safest way to achieve conscious sedation in patients with intellectual disability. Since defining the effectiveness of sedation solely based on the success of dental treatment is reductive, the present study was planned to provide clinicians with an overall figure for conscious sedation procedures in individuals with intellectual disabilities, considering not only the success of dental treatment but also the number of side effects and the achievement of an adequate level of sedation.

Nitrous oxide proved to be the most effective drug to perform conscious sedation in subjects with intellectual disability. Even if the use of conscious sedation has been described since the 80s [30], few studies have been conducted to date to investigate its use in these special patients, compared to the large amount of literature on healthy children [39]. Only four studies were based exclusively on individuals under the age of 18 with intellectual disabilities, and among these, there is great heterogeneity in both the drugs used and the route of administration [17,32,36,37].

The treatment of these patients requires more time, effort, equipment, and energy to ensure the same quantity and quality of dental care provided to general patients [40]. The majority of the research conducted on subjects with disability focuses on the use of general anesthesia in order to achieve dental treatment [41]. General anesthesia is described as the most effective modality for providing dental care to patients who have difficulty accepting treatment. However, GA is the most complex and expensive procedure to arrange and has the greatest risk of side effects [40]. Whereas conscious sedation via enteral benzodiazepines or parenteral benzodiazepines or nitrous oxide or a mixed technique, was found to be an effective method of performing dental treatment with a success rate of more than 75% and the success for nitrous oxide alone was even higher, as reported in 1259 sedations. Parenteral benzodiazepines showed the lowest number of adverse events, while nitrous oxide and enteral benzodiazepines showed similar results with less than 20% of events. The most frequently described adverse events were respiratory problems, nausea and vomiting, neurological effects, vasovagal symptoms, behavioral alterations, and paradoxical effects. Nitrous oxide and enteral benzodiazepines have proven to be the safest drugs, with less than 2% deep sedation in more than a thousand subjects. This percentage increases significantly when nitrous oxide is used in addition to oral benzodiazepines.

Dental treatment under general anesthesia is not repeatable at close intervals, is not aimed at increasing patient cooperation, and is often not followed by a program of recall visits [42]. Conversely, with conscious sedation, the patient may cooperate during the treatment, have a memory of the procedure, and more importantly, have the possibility of decreasing sedation levels for future treatments. Repeated sessions of conscious sedation have indeed been shown to significantly improve the level of cooperation [26]. These advantages may enable some patients, even some of those with intellectual disabilities, to cope with dental treatment without resorting to sedative drugs [11]. It is therefore of primary importance that patients with intellectual impairment have the opportunity to enter a dedicated program with frequent dental referrals. Indeed, it has been shown that patients who learn to cooperate in an oral examination will successfully maintain the same behavior in future visits. In addition, they may also acquire skills in some more demanding procedures [43].

Limitations of this review include the high heterogeneity of the studies analyzed in terms of drugs used, mode of administration, age range of the samples, and factors that may influence the success of therapy. Furthermore, given the limited number of studies available, it was not possible to exclude older studies or those with a medium or high risk of bias. Further standardized studies are needed to strengthen recommendations on the choice of drugs for performing dental treatment under conscious sedation in patients with intellectual disabilities, although current data seem to suggest this pharmacological approach as an effective and safe strategy to facilitate dental treatments.

The strengths of the present study are the large number of subjects included and the innovative method of pooling data to provide a success rate that takes into account not only the completion of treatment but also the safety of each drug and the mode of administration. These aspects are important for all patients, but crucial for those with special needs.

## 5. Conclusions

Considering the low risk of side effects and deep sedation, nitrous oxide appears to be the best choice to perform conscious sedation in individuals with intellectual disability undergoing dental treatment. This should be the first approach, which, in case of failure, can be supplemented with oral benzodiazepines.

Given the possibility of using safe and effective sedation techniques, general anesthesia should only be reserved for the most severe cases in which the described techniques fail or cannot be performed due to patients’ characteristics. The strength of the recommendation remains low due to the great heterogeneity of the included studies, their design, and the risk of bias found.

## Figures and Tables

**Figure 1 ijerph-20-01779-f001:**
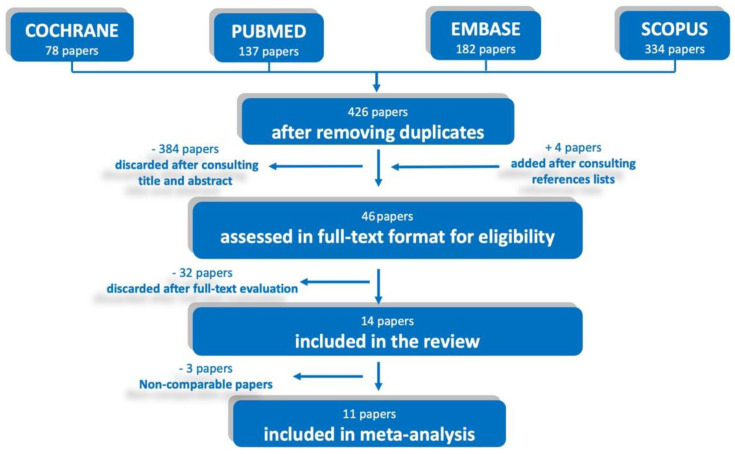
PRISMA flow chart of the search process.

**Figure 2 ijerph-20-01779-f002:**
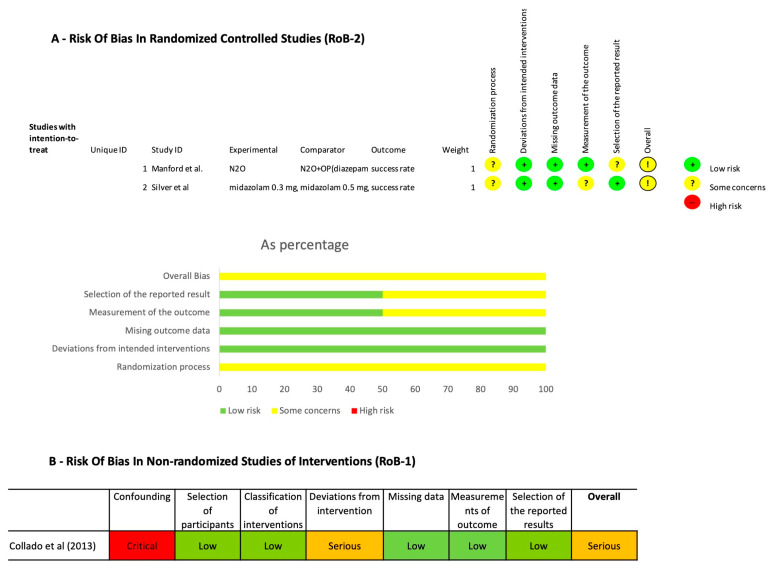
Risk of bias of RCTs using RoB 2 tool (**A**) and NRCT using Robins 1 (**B**).

**Figure 3 ijerph-20-01779-f003:**
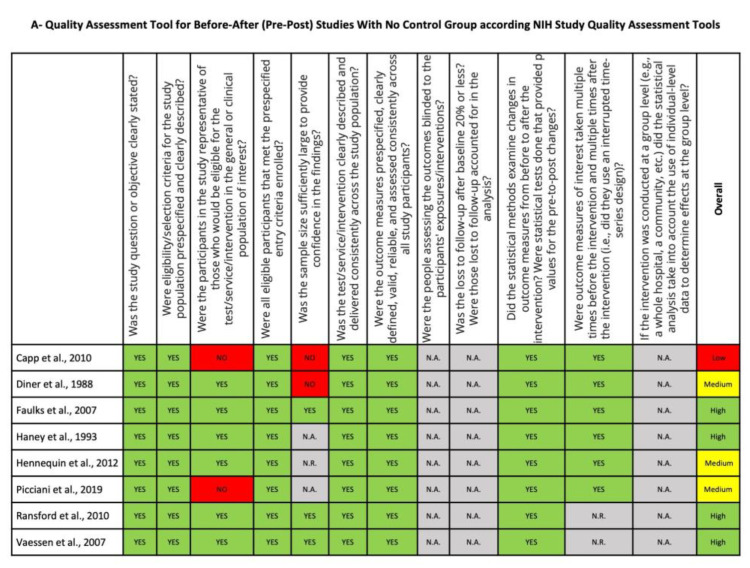
Quality assessment of before-after studies with no control group (**A**) and for observational cohort and cross-sectional studies (**B**) using NIH study quality assessment tools.

**Figure 4 ijerph-20-01779-f004:**
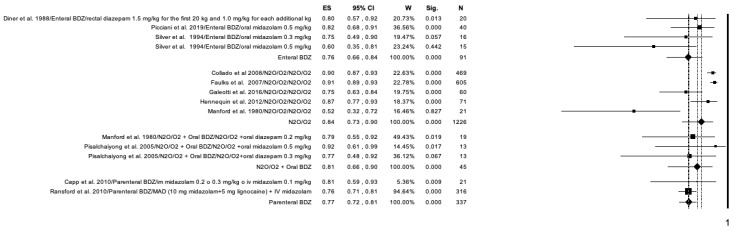
Forrest plot for success rate of sedation in the included studies. Results according to drug used and administration modality.

**Figure 5 ijerph-20-01779-f005:**
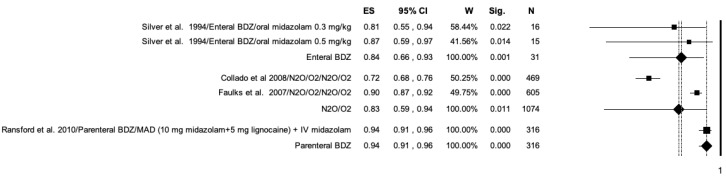
Forest plot for the occurrence of adverse events due to sedation in the included studies. Results according to drug used and administration modality.

**Figure 6 ijerph-20-01779-f006:**
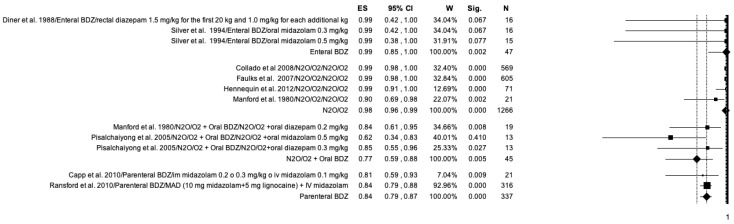
Forest plot for the absence of deep sedation in the included studies. Results according to drug used and administration modality.

**Figure 7 ijerph-20-01779-f007:**
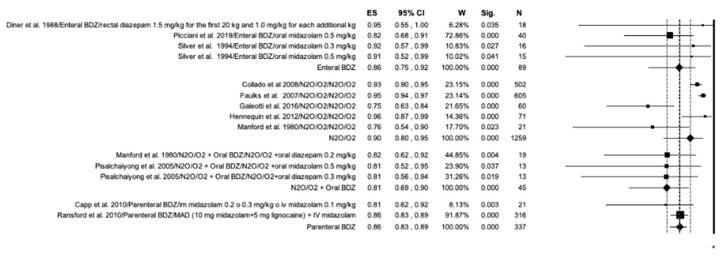
Forest plot for the overall efficiency of the included studies obtained combined the success rate, the non-occurrence of side effects and the absent of deep sedation. Results according to drug used and administration modality.

**Table 1 ijerph-20-01779-t001:** General characteristics of the studies included.

Authors	Year	Journal	Country	Databases	Type of Study
Capp et al. [33]	2010	European Journal of Paediatric Dentistry	Brazil	PMD, SC, EMB	Prospective clinical study
Collado et al. [28]	2008	BMC Clinical Pharmacology	France	PMD, SC, EMB	Multicenter Prospective clinical study
Collado et al. [26]	2013	Plos One	France	Reference	NRCT
Diner et al. [35]	1988	Special Care in Dentistry	Canada	SC, EMB	Prospective clinical study
Faulks et al. [29]	2007	Developmental Medicine and Child Neurology	France	PMD, SC, EMB	Prospective clinical study
Galeotti et al. [36]	2016	BioMed Research International	Italy	PMD, SC, EMB	Observational study
Haney et al. [17]	1993	ASDC Journal of Dentistry for Children	USA	SC, EMB	Retrospective study
Hennequin et al. [27]	2012	Clinical Oral Investigations	France	SC, EMB	Multicenter Prospective clinical study
Manford et al. [30]	1980	Anaesthesia	UK	CHR, SC, EMB	Randomized control trial
Picciani et al. [34]	2019	Journal of Clinical and Experimental Dentistry	Brazil	SC, EMB	Prospective clinical study
Pisalchaiyong et al. [37]	2005	Pediatric Dentistry	Thailand	PMD, SC	Prospective randomized, cross-over study
Ransford et al. [31]	2010	British Dental Journal	United Kingdom	SC, EMB	Multicenter Prospective clinical study
Silver et al. [32]	1994	Pediatric Dentistry	UK	CHR, SC,	Randomized control trial
Vaessen et al. [18]	2017	Special Care in Dentistry	The Netherlands	SC, EMB	Retrospective study

PMD (PubMed); SC (Scopus); EMB (Embase); CHR (Cochrane Central Register of Controlled Trials); RCT (randomized control trial); NRCT (non-randomized control trial).

**Table 2 ijerph-20-01779-t002:** Main characteristics of the studies that used nitrous oxide alone or in combination with other drugs.

Author (Years)	N of Patients	Mean Age (Range)	Evaluation Scale	Administered Drug/Operator	Results of Primary Outcome		Results of Secondary Outcome	
	N of Sedations	Sex (M/F)		Dental Procedures				
Galeotti et al. (2016) [36]	472	6.6	Modified Venham Scale, Vital signs	N_2_O/O_2_ (at different concentrations) by dentist	Sedation efficacy:	75%	Behavior assessment: n.a.	
	(60 *)	(4–17)			Side effects:	n.a.	Vital signs: n.a.	
					Deep sedation:	n.a.		
	472	n.a.		Oral examination, oral hygiene, restorative treatment, oral surgery				
	(60 *)							

Collado et al. (2013) [26]	142	30.5	Venham Scale, Ramsay score for sedation assessment, Vital signs	iv midazolam (8.8 mg +/-4.9 mg) and N_2_O/O_2_ (50/50%) if necessary, premedication with Midazolam (os/ra 0.3 to 0.5 mg/kg), if necessary, by dentist	Sedation efficacy:	89%	Behavior assessment (Venham score 0):	
	(98 *)	(8–57)			Side effects:	16%	- at venous cannulation = 23%	
					Deep sedation:	3%	- at the end of the induction = 59%	
	320	113/74					- during dental treatment = 40%	
	(187 *)						Vital signs:	
							- SpO_2_ < 90 = 16% of patients	
							- HR < minimal normative value in relation to the age, <1%	
				Oral examination, radiographs, impressions, scaling, restorative treatment, prosthetic treatment, oral surgery			- SBP < minimal normative value, <1%	
							- SBP > maximal normative value 34%	
							- DBP < minimal normative value, 1%	
							- DBP > maximal normative value, <1%	
							Influence of repeated sedation:	
							Venham score 0 decreased at venous cannulation (*p* = 0.01), and during dental treatment (*p* < 0.01)	
							Level of sedation:	
							During treatment Ramsey score = 1.96 (±0.72)	
Hennequin et al. (2012) [27]	549	22.8	Venham Scale, VAS Scale	N_2_O/O_2_ (50/50%) by dentist	Sedation efficacy:	87%	Behavior assessment: n.a.	
	(n.a.*)	(1–80)			Side effects:	n.a.	Patient/dentist satisfaction: n.a.	
				Oral examination, scaling, restorative treatment, oral surgery	Deep sedation:	0%		
	638	308/241						
	(71*)							
Collado et al. (2008) [28]	662	n.a.	Venham Scale	N_2_O/O_2_ (50/50%) by dentist	Sedation efficacy:	90%	Behavior assessment:	
	(325 *)	(>5)			Side effects:	28%	Cooperation increased from application of the mask to perioperative steps (*p* < 0.01)	
				Oral examination, radiograph, oral hygiene, restorative treatment, oral surgery	Deep sedation:	0%		
	826	n.a.					Role of operator on success:	
	(469 *)						Not-expert vs experts (failures 13% vs 9% (*p* < 0.01)	
Faulks et al. (2007) [29]	349	22	Venham Scale	N_2_O/O_2_ (50/50%) by dentist	Sedation efficacy:	91%	Behavior assessment:	
		(3–81)			Side effects:		Venham scores: decrease from mask application to treatment performance (*p* < 0.01), and during local anesthesia (*p* < 0.01). Autistic patients showed poorer cooperation compared to other IDs (*p* < 0.01)	
	605	192/157		n.a.	10%; nausea/vomiting > in longer sedation (*p* < 0.01)			

					Deep sedation:	0%		
Pisalchaiyong et al. (2005) [37]	13	8.7	Rating scale for sleep, body movement and crying behavior	N_2_O/O_2_ (50/50%) plus diazepam (0.3 mg/kg) or N_2_O/O_2_ (50/50%) plus midazolam (0.5 mg/kg) by dentist and anesthesiologist	Sedation efficacy:		Behavior assessment:	
		(5–15)			- 77% N_2_O/O_2_ plus Diazepam;		77% who received diazepam and 100% who received midazolam rated as “good” and “very good”	
					- 92% N_2_O/O_2_ plus Midazolam			
	26	10/3						
					Side effects: n.a.			
				Preventive procedures, scaling, restorative treatment, prosthetics treatment, oral surgery	Deep sedation			
					- 5% N_2_O/O_2_ plus Diazepam			
					- 38% N_2_O/O_2_ plus Midazolam			
Haney et al. (1993) [17]	143	6.4	n.a.	Meperidine (1mg/lb) plus promethazine (0,5mg/lb) plus N_2_O/O_2_ (≤ 50/50%) by dentist	Sedation efficacy:	68%	Role of operator on success: n.a.	
	(* n.a.)	(2–18)			Side effects:	3%	Level of sedation:	
							- 32% failure of sedation	
							- 34% moderate success	
	282	n.a.					- 34% excellent success	
	(120 *)			n.a.				
Manford et al. (1980) [30]	40	n.a.	Customized behavioral scale	N_2_O/O_2_ or N_2_O/O_2_ plus iv diazepam (0.2 mg/kg) by dentist and anesthesiologist	Sedation efficacy:		Behavior assessment: n.a. (reported in graphs)	
		(5–22)			- 52% N_2_O_2_		Treatment acceptance: n.a.	
					- 79% N_2_O_2_ plus iv diazepam			
	40	n.a.						
				n.a.	Deep sedation:			
					- 10% N_2_O_2_			
					- 16% N_2_O_2_ plusand iv diazepam			

ID (intellectual disability); n.a. (data not available); n.s. (not significant); im (intramuscular drug administration); in (intranasal drug administration); iv (intravenous drug administration); MAD (mucosal atomization device); os (oral drug administration); ra (rectal drug administration); TPC (target plasma concentration); DBP (diastolic blood pressure); SBP (systolic blood pressure); HR (heart rate); SpO_2_ (oxygen saturation); * for the metanalysis, only patients with intellectual disabilities were considered.

**Table 3 ijerph-20-01779-t003:** Main characteristics of the studies that did not use nitrous oxide alone or in combination with other drugs.

Author (Years)	N of Patients	Mean Age (Range)	Evaluation Scale	Administered Drug/Operator	Results of Primary Outcome		Results of Secondary Outcome	
	N of Sedations	Sex (M/F)		Dental Procedures				
Picciani et al. (2019) [34]	40	18	Vital signs	Midazolam (OS 0.5 mg/kg) by dentist and anesthesiologist	Sedation efficacy:	82%	Vital signs:	
		(6–73)					- Decrease BP and HR (*p* < 0.01)	
							- O_2_ > 97% (n.s.)	
	40	28/12						
				Preventive procedure, impression, restorative treatment, oral surgery				


Vaessen et al. (2017) [18]	124	52	OAA/S, vital signs	Propofol (1%, TPC 1.5 μg ml) by dentist and anesthesiologist	Sedation efficacy:	100%	Level of sedation:	
		(18–75)			Side effects:	37%	OAA/S: 4.1	
					Deep sedation:	27%	Vital signs: n.a.	
	124	n.a.						
				Oral examination, radiograph, scaling, restorative treatment, oral surgery				



Capp et al. (2010) [33]	40	n.a.	Customized behavioral scale (A = allowed treatment, B = reacted to stimuli but allowed treatment, C = not allowed treatment)	Midazolam (im 0.2–0.3 mg/kg or iv 0.1 mg/kg) by dentist and anesthesiologist	Sedation efficacy:	81%	Behavior assessment:	
	(21 *)	(2–54)			Deep sedation:	19%	- A = 48%	
							- B = 33%	
	40	n.a.					- C = 19%	
	(21*)			Restorative treatment, oral surgery				

Ransford et al. (2010) [31]	289	n.a.	Dental sedation teachers group scale for behavior and level of sedation; acceptability of treatment, vital signs	Midazolam (MAD or iv) by dentist and anesthesiologist	Sedation efficacy:	76%	Treatment acceptance:	
		(>18)			Side effects:	6%	- “Very good” or “good” = 90%	
					Deep sedation:	17%.	- “Much better” or “better” than GA = 87%	
	316	n.a.		n.a.			- “Much better” or “better” than OS = 94%	
							Behavior assessment:	
							- Good = 50%	
							- Fair = 29%	
							- Poor = 16%	
							- Very poor = 5%	
							Level of sedation:	
							- Fully awake = 12%	
							- Drowsy = 41%	
							- Verbal responsive = 30%	
							- Physical responsive = 16%	
							- Unresponsive = 1%	
							Vital signs: n.a.	
Silver et al. (1994) [32]	31	9	Customized behavior scale (modified Frankl Scale deleting G4), vital signs	Midazolam (os 0.3 versus 0.5 mg/kg) by dentist and anesthesiologist	Sedation efficacy:		Behavior assessment:	
		(3–18)			- 0.3 mg/kg 75%		T1 score 3	T2 score 3
					- 0.5 mg/kg 60%		- GA: 44%,	- GA 50%
	31	16/15			Side effects:		- GB: 27%	- GB 33.3%
				n.a.	- 0.3 mg/kg 6%		Vital signs: n.a. (reported in graphs)	
					- 0.5 mg/kg 0%			
					Deep sedation:	0%		
Diner et al. (1988) [35]	42	n.a.	Customized behavioral scale (evaluation of movements of head, arms, trunk, legs), vital signs	Diazepam (ra 1.5 mg/kg for the first 20 kg of weight + 1 mg/kg for additional kg) by dentist, hygienist and anesthesiologist	Sedation efficacy:	80%	Behavior assessment:	
		(4–31)			Deep sedation:	0%	Improved = 80%	
							Unchanged = 15%	
	20	n.a.					Worsened = 5%	
							Vital signs: n.a.	

				Preventive procedure, scaling				

ID (intellectual disability); n.a. (data not available); n.s. (not significant); im (intramuscular drug administration); in (intranasal drug administration); iv (intravenous drug administration); MAD (mucosal atomization device); os (oral drug administration); ra (rectal drug administration); TPC (target plasma concentration); DBP (diastolic blood pressure); SBP (systolic blood pressure); HR (heart rate); SpO_2_ (oxygen saturation). * For the metanalysis, only patients with intellectual disabilities were considered.

## Data Availability

Not applicable.

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
