# Peer review of "Conscious Sedation for Dental Treatments in Subjects with Intellectual Disability: A Systematic Review and Meta-Analysis"

_ijerph, 2023, doi:10.3390/ijerph20031779_

Round 1

Reviewer 1 Report

The manuscript is entitiled "Conscious Sedation for Dental Treatments in Subjects with In-2 tellectual Disability. A Systematic Review and Meta-Analysis"

The authors attempted to investigate the efficacy of conscious sedation in patients with intellectual disabilities undergoing dental treatment.

I have some minor comments-

In the table 2 & 3, authors presented many data which is not available (n.a.). The authors should add an explanation of why they included the study when all the information was not available.

The quality of figures 2 and 3 is poor. The authors should amend the figures.

Line 120, change N2O to N2O.

The authors followed the proper guidelines for a systematic review, and the manuscript is well written and has significant importance. Therefore, the manuscript can be accepted.

Author Response

We would like to thank the reviewer for the hints provided to improve our paper.

Reviewer 1

The manuscript is entitiled "Conscious Sedation for Dental Treatments in Subjects with Intellectual Disability. A Systematic Review and Meta-Analysis"

The authors attempted to investigate the efficacy of conscious sedation in patients with intellectual disabilities undergoing dental treatment.

I have some minor comments-

In the table 2 & 3, authors presented many data which is not available (n.a.). The authors should add an explanation of why they included the study when all the information was not available.

An explanation was added to M&M with the following sentences, the first added to “Data collection and synthesis” and the second to “Outcome variables”

  • Studies whose sample did not include only patients with intellectual disabilities were included in the study only if it was possible to extrapolate data for subjects with intellectual disabilities for at least one primary outcome.
  • Data were reported only if they were related to subjects with intellectual disability. in studies whose sample did not include only patients with intellectual disability where it was not possible to extrapolate data related to the subgroup under study, numerical data were not reported.

The quality of figures 2 and 3 is poor. The authors should amend the figures.

The figures were modified

Line 120, change N2O to N2O.

It was changed

The authors followed the proper guidelines for a systematic review, and the manuscript is well written and has significant importance. Therefore, the manuscript can be accepted.

Thank you for appreciating our work

Reviewer 2 Report

 The authors investigated the efficiency of various types of conscious sedation in intellectually disabled patients with a systematic review and meta-analysis and concluded that nitrous oxide is the best option to perform conscious sedation for them. This study is interesting and clinically meaningful when selecting sedative drugs. However, some concerns should be addressed before publication.

           Major comment

General comment; As the authors mentioned in the discussion, the studies included for analysis are heterogeneous in terms of dental treatments and patients’ characteristics. To compensate for the heterogeneity of effects among studies, the authors utilized different statistical approaches such as the random effect model. However, despite the use of sophisticated statistical methods, questions arise about whether the heterogeneity settles down with these methods. For example, the severity of intellectual disability may be related to the success rate of sedation and adverse outcomes since they have a high probability of poor behaviors requiring high doses of sedatives. Although the approach employed in this study makes up the limitation of each study, it should be more clearly stated how the statistical analysis reduces the confounding bias caused by the differences in the severity of ID, dental treatment types, treatment time, the dose of sedative, etc among the studies.

Minor comment

1.     Please describe the criteria of the studies to be included in the review in more detail. For example, it remains unclear why 32 papers were discarded after full text evaluation during the screening.

2.     Fig2-7; It is hard to read the figures. These should be modified to be readable. 

Author Response

The reviewer made some interesting comments.

The authors investigated the efficiency of various types of conscious sedation in intellectually disabled patients with a systematic review and meta-analysis and concluded that nitrous oxide is the best option to perform conscious sedation for them. This study is interesting and clinically meaningful when selecting sedative drugs. However, some concerns should be addressed before publication.

Major comment

General comment; As the authors mentioned in the discussion, the studies included for analysis are heterogeneous in terms of dental treatments and patients’ characteristics. To compensate for the heterogeneity of effects among studies, the authors utilized different statistical approaches such as the random effect model. However, despite the use of sophisticated statistical methods, questions arise about whether the heterogeneity settles down with these methods. For example, the severity of intellectual disability may be related to the success rate of sedation and adverse outcomes since they have a high probability of poor behaviors requiring high doses of sedatives. Although the approach employed in this study makes up the limitation of each study, it should be more clearly stated how the statistical analysis reduces the confounding bias caused by the differences in the severity of ID, dental treatment types, treatment time, the dose of sedative, etc among the studies.

The reviewer went straight to the most elusive points of the paper, the heterogeneity of the studies included. As is reported in the Cochrane Handbook chapter 10 “It is clearly of interest to determine the causes of heterogeneity among results of studies. This process is problematic since there are often many characteristics that vary across studies from which one may choose. Heterogeneity may be explored by conducting subgroup analyses”. The included papers of the present systematic revision had high heterogeneity in all aspects (i.e. type of study, data collected, ID, dental treatment types, treatment time, the dose of sedative, etc). We followed the suggestion of the Cochrane Handbook, still the heterogeneity is definitively high. The only other solution (also proposed Ironically in the Cochrane Handbook) is not performed a meta-analysis and ignore heterogeneity and get no conclusion at all.

Minor comment

  1. Please describe the criteria of the studies to be included in the review in more detail. For example, it remains unclear why 32 papers were discarded after full text evaluation during the screening.

The reasons for papers’ exclusion at each step of selection are reported in Supplementary files (S3 and S5). The titles of the two supplementary files were modified to better clarify.

  1. Fig2-7; It is hard to read the figures. These should be modified to be readable.

Figures were modified

Reviewer 3 Report

Dear authors, thanks for the interesting article. I have a few questions about the content

1. Why were only articles in English, Italian, and French included in the review? We know that thanks to the activities of, for example, EFAAD, namely a group of scientists from Germany, numerous convincing results in variable treatment regimens have been obtained in the field of Conscious Sedation for Dental Treatments. Suffice it to recall the scientific school under the guidance of Professor Jacobs

2. I would also like to clarify the legal issue of the status of "Conscious Sedation", since, depending on the legislative framework, both dentists and anesthesiologists can perform this type of treatment. Accordingly, the scientific and practical significance of the methodology can be assessed in different ways, taking into account the profile of a specialist.

3. Why was the article on Xenon inhalant not evaluated?

Author Response

Thanks a lot for the nice comments from the reviewer.

Dear authors, thanks for the interesting article. I have a few questions about the content

  1. Why were only articles in English, Italian, and French included in the review? We know that thanks to the activities of, for example, EFAAD, namely a group of scientists from Germany, numerous convincing results in variable treatment regimens have been obtained in the field of Conscious Sedation for Dental Treatments. Suffice it to recall the scientific school under the guidance of Professor Jacobs

A pre-assessment of the studies (not reported in the article) was carried out to assess the availability of articles in languages other than English, without finding substantial results. However, Italian and French were taken into account as mother tongues of the authors.

With regard to Prof. F.E. Jacobs' studies, only one article was found that was excluded because it was a case report and did not have an abstract. If, on the other hand, the reviewer referred to the studies of Prof. W. Jakobs, only one paper was found which was excluded because it described a sedative method without reporting data on its success.

  1. I would also like to clarify the legal issue of the status of "Conscious Sedation", since, depending on the legislative framework, both dentists and anesthesiologists can perform this type of treatment. Accordingly, the scientific and practical significance of the methodology can be assessed in different ways, taking into account the profile of a specialist.

The authors agree with this statement. That is why it was deemed necessary to specify in the introduction section the definition of conscious sedation and the possibility both for the dentist and anesthetist to perform it. Furthermore, the operator who performed the sedation in the included studies was in most of them a dentist (13 papers out of 14). Therefore, it was not possible to analyze the impact of the operator on the outcomes.

  1. Why was the article on Xenon inhalant not evaluated?

Research with the strings reported in the study produced no results on the use of xenon to perform conscious sedation in dentistry. A new search was performed on pubmed, embase and scopus with the following keywords "xenon" and "sedation" and "dentistry" and produced no results except for two articles in Russian.